# The landscape of antibody binding in SARS-CoV-2 infection

Anna S. Heffron[1], Sean J. McIlwain[2,3], Maya F. Amjadi[4], David A. Baker[1], Saniya Khullar[2], Tammy Armbrust[5], Peter J. Halfmann[5], Yoshihiro Kawaoka[5], Ajay K. Sethi[6], Ann C. Palmenberg[7], Miriam A. Shelef[4,8], David H. O'Connor[1,9], Irene M. Ong[2,3,10,11] *

1 Department of Pathology and Laboratory Medicine, University of Wisconsin-Madison, Madison, Wisconsin, United States of America, 2 Department of Biostatistics and Medical Informatics, University of Wisconsin-Madison, Madison, Wisconsin, United States of America, 3 University of Wisconsin Carbone Comprehensive Cancer Center, University of Wisconsin-Madison, Madison, Wisconsin, United States of America, 4 Department of Medicine, University of Wisconsin-Madison, Madison, Wisconsin, United States of America, 5 Department of Pathobiological Sciences, School of Veterinary Medicine, Influenza Research Institute, University of Wisconsin-Madison, Madison, Wisconsin, United States of America, 6 Department of Population Health Sciences, University of Wisconsin-Madison, Madison, Wisconsin, United States of America, 7 Department of Biochemistry, Institute for Molecular Virology, University of Wisconsin–Madison, Madison, Wisconsin, United States of America, 8 William S. Middleton Memorial Veterans Hospital, Madison, Wisconsin, United States of America, 9 Wisconsin National Primate Research Center, University of Wisconsin-Madison, Madison, Wisconsin, United States of America, 10 Department of Obstetrics and Gynecology, University of Wisconsin-Madison, Madison, Wisconsin, United States of America, 11 Center for Human Genomics and Precision Medicine, University of Wisconsin-Madison, Madison, Wisconsin, United States of America

* irene.ong@wisc.edu

**Data Availability Statement:** All peptide microarray datasets and code used in these analyses can be downloaded from https://github.com/Ong-Research/UW_Adult_Covid-19.

## Abstract

The search for potential antibody-based diagnostics, vaccines, and therapeutics for pandemic severe acute respiratory syndrome coronavirus 2 (SARS-CoV-2) has focused almost exclusively on the spike (S) and nucleocapsid (N) proteins. Coronavirus membrane (M), ORF3a, and ORF8 proteins are humoral immunogens in other coronaviruses (CoVs) but remain largely uninvestigated for SARS-CoV-2. Here, we use ultradense peptide microarray mapping to show that SARS-CoV-2 infection induces robust antibody responses to epitopes throughout the SARS-CoV-2 proteome, particularly in M, in which 1 epitope achieved excellent diagnostic accuracy. We map 79 B cell epitopes throughout the SARS-CoV-2 proteome and demonstrate that antibodies that develop in response to SARS-CoV-2 infection bind homologous peptide sequences in the 6 other known human CoVs. We also confirm reactivity against 4 of our top-ranking epitopes by enzyme-linked immunosorbent assay (ELISA). Illness severity correlated with increased reactivity to 9 SARS-CoV-2 epitopes in S, M, N, and ORF3a in our population. Our results demonstrate previously unknown, highly reactive B cell epitopes throughout the full proteome of SARS-CoV-2 and other CoV proteins.

## Introduction

Antibodies correlate with protection from coronaviruses (CoVs) including severe acute respiratory syndrome coronavirus 2 (SARS-CoV-2) [1–8], severe acute respiratory syndrome coronavirus (SARS-CoV) [8–12], and Middle Eastern respiratory syndrome coronavirus

**Funding:** I.M.O. acknowledges support by the Clinical and Translational Science Award (CTSA) program (ncats.nih.gov/ctsa), through the National Institutes of Health National Center for Advancing Translational Sciences (NCATS), grants UL1TR002373 and KL2TR002374. This research was also supported by 2U19AI104317-06 (to I.M.O via James Gern) and R24OD017850 (to D.H.O.) from the National Institute of Allergy and Infectious Diseases of the National Institutes of Health (www.niaid.nih.gov). A.S.H. has been supported by the National Institutes of Health National Research Service Award T32 AI007414 and M.F.A. by T32 AG000213 (www.nlm.nih.gov/ep/NRSAFellowshipGrants.html). S.J.M. acknowledges support by the National Cancer Institute, National Institutes of Health and University of Wisconsin Carbone Comprehensive Cancer Center's Cancer Informatics Shared Resource (grant P30-CA-14520; cancer.wisc.edu/research/) and by the National Institute of Allergy and Infectious Diseases of the National Institutes of Health 2U19AI104317-06. This project was also funded through a COVID-19 Response Grant from the Wisconsin Partnership Program and the University of Wisconsin School of Medicine and Public Health (to M.A.S.; www.med.wisc.edu/wisconsin-partnership-program/), startup funds through the University of Wisconsin Department of Obstetrics and Gynecology (I.M.O.; www.obgyn.wisc.edu/), and the Data Science Initiative (research.wisc.edu/funding/data-science-initiative/) grant from the University of Wisconsin-Madison Office of the Chancellor and the Vice Chancellor for Research and Graduate Education (with funding from the Wisconsin Alumni Research Foundation) (I.M.O.). The funders had no role in study design, data collection and analysis, decision to publish, or preparation of the manuscript.

**Competing interests:** I have read the journal's policy and the authors of this manuscript have the following competing interests: The authors declare the following competing interests: A.S.H., S.J.M., D.A.B., M.F.A., S.K., M.A.S., D.H.O., and I.M.O are listed as the inventors on a patent filed that is related to findings in this study. Application: 63/080568, 63/083671. Title: IDENTIFICATION OF SARS-COV-2 EPITOPES DISCRIMINATING COVID-19 INFECTION FROM CONTROL AND METHODS OF USE. Application type: Provisional. Status: Filed. Country: United States. Filing date: September 18, 2020, September 25, 2020.

**Abbreviations:** ACE2, angiotensin converting enzyme 2; β-CoV, betacoronavirus; BH, Benjamini–Hochberg; CCCoVs, "common cold" CoVs; CoV, coronavirus; COVID-19, coronavirus disease 2019;

(MERS-CoV) [8,13–16]. All CoVs encode 4 main structural proteins, spike (S), envelope (E), membrane (M), and nucleocapsid (N), as well as multiple nonstructural proteins and accessory proteins [17]. In SARS-CoV-2, anti-S and anti-N antibodies have received the most attention to date [1–8], including in serology-based diagnostic tests [1–5] and vaccine candidates [6–8]. The immunogenicity of S-based vaccines is variable [18,19], so better representation of the breadth of antibody reactivity in vaccines, therapeutics, and diagnostics will be important as the pandemic continues especially as new variants emerge. Prior reports observed that not all individuals infected with SARS-CoV-2 produce detectable antibodies against S or N [1–5], indicating a need for expanded antibody-based options.

Much less is known about antibody responses to other SARS-CoV-2 proteins, although data from other CoVs suggest they may be important. Antibodies against SARS-CoV M can be more potent than antibodies against SARS-CoV S [20–22], and some experimental SARS-CoV and MERS-CoV vaccines elicit responses to M, E, and ORF8 [8]. Additionally, previous work has demonstrated humoral cross-reactivity between CoVs [7,11,23–26] and suggested it could be protective [26,27], although full-proteome cross-reactivity has not been investigated.

We designed a peptide microarray tiling the proteomes of SARS-CoV-2 and 8 other human and animal CoVs in order to assess antibody epitope specificity and potential cross-reactivity with other CoVs. We examined immunoglobulin G (IgG) antibody responses in 40 coronavirus disease 2019 (COVID-19) convalescent patients and 20 SARS-CoV-2-naïve controls. Independent enzyme-linked immunosorbent assays (ELISAs) confirm 4 of the highest-performing epitopes. We detected antibody responses to epitopes throughout the SARS-CoV-2 proteome, with several antibodies exhibiting apparent cross-reactive binding to homologous epitopes in multiple other CoVs.

## Results

### SARS-CoV-2-naïve controls show consistent binding in "common cold" CoVs and limited binding in SARS-CoV-2, SARS-CoV, and MERS-CoV

Greater than 90% of adult humans are seropositive for the human "common cold" CoVs (CCCoVs: HCoV-HKU1, HCoV-OC43, HCoV-NL63, and HCoV-229E) [28,29], but the effect of these preexisting antibodies upon immune responses to SARS-CoV-2 or other CoVs remains uncertain. We measured IgG reactivity in sera from 20 SARS-CoV-2-naïve controls to CoV linear peptides, considering reactivity that was >3.00 standard deviations above the mean for the log$_2$-quantile normalized array data to be indicative of antibody binding [30]. All sera (SARS-CoV-2-naïve and COVID-19-convalescent) exhibited binding in known epitopes of at least one of the control non-CoV strains (poliovirus vaccine and rhinovirus; Fig 1, S1 Data), and all were collected in Wisconsin, United States of America, where exposure to SARS-CoV or MERS-CoV was extremely unlikely. We found that at least 1 epitope in structural or accessory proteins showed binding in 100% of controls for HCoV-HKU1, 85% of controls for HCoV-OC43, 65% for HCoV-NL63, and 55% for HCoV-229E (Fig 2, S2 Data). The apparent cross-reactive binding was observed in 45% of controls for MERS-CoV, 50% for SARS-CoV, and 50% for SARS-CoV-2 (S2 Data). We completed neutralization assays on 12 of these control samples and on 18 additional samples from other SARS-CoV-2-naïve controls collected before 2019, and none of these had detectable neutralization activity against SARS-CoV-2 [31] (S1 Table, S1 Fig).

### SARS-CoV-2 infection induces antibodies binding throughout the proteome

We aimed to map the full breadth of IgG binding induced by SARS-CoV-2 infection and to rank the identified epitopes in terms of likelihood of immunodominance. We defined epitope

E, envelope; ELISA, enzyme-linked immunosorbent assay; EMR, electronic medical record; IgG, immunoglobulin G; M, membrane; MAS, Maskless Array Synthesizer; MERS-CoV, Middle Eastern respiratory syndrome coronavirus; N, nucleocapsid; NPPOC, 2-(2-nitrophenyl) propyloxycarbonyl; PBS, phosphate-buffered saline; RBD, receptor-binding domain; S, spike; SARS-CoV, severe acute respiratory syndrome coronavirus; SARS-CoV-2, severe acute respiratory syndrome coronavirus 2; TBS, tris-buffered saline; UW, University of Wisconsin; VOC, variant of concern.

recognition as antibody binding to contiguous peptides in which the average $\log_2$-normalized intensity for patients was at least 2-fold greater than for controls with $t$ test statistics yielding adjusted $p$-values <0.1. We chose these criteria, rather than the 3.00 standard deviation cutoff (S2 Data), in order to ensure that binding detected would be greater than background binding seen in controls (2-fold greater) and to remove regions of binding that were not at least weakly significantly different from controls (adjusted $p < 0.1$). All COVID-19 convalescent patients' sera bound multiple epitopes in SARS-CoV-2, including in 2 patients who did not have detectable neutralizing antibodies in neutralization assays (S1 Table, S1 Fig). Top-ranking epitopes had greater correlations (0.7 and greater) with neutralization titers (Table 1).

These criteria identified 79 B cell epitopes (Fig 3, Table 1) in S, M, N, ORF1ab, ORF3a, ORF6, and ORF8. We ranked these epitopes by minimum adjusted $p$-value for any 16-mer in the epitope in order to determine the greatest likelihood of difference from controls as a proxy for likelihood of immunodominance. The highest-ranking epitope occurred in the N-terminus of M (1-M-24). Patient sera showed high-magnitude reactivity (up to an average of 6.7 fluorescence intensity units) in other epitopes in S, M, N, and ORF3a, with lower-magnitude reactivity (average of <3.3 fluorescence intensity units) epitopes in other proteins. The epitopes with the greatest reactivity in S were located in the S2 subunit of the protein (residues 686–1273) rather than the S1 subunit (residues 14–685) [6] (Fig 3). The greatest reactivity in S occurred in the fusion peptide (residues 788–806) and at the base of the extracellular portion of the protein (between the heptad repeat 1 and heptad repeat 2, roughly residues 984–1163) (Figs 3 and 4). The highest magnitude antibody binding (red sites in Fig 4A) on S are below the flexible head region that must be in the "up" position for angiotensin converting enzyme 2 (ACE2) binding to occur. Notably less reactivity occurred in the receptor-binding domain (RBD) (residues 319–541) [6]. Four detected epitopes (553-S-26, 624-S-23, 807-S-26, and 1140-S-25) have

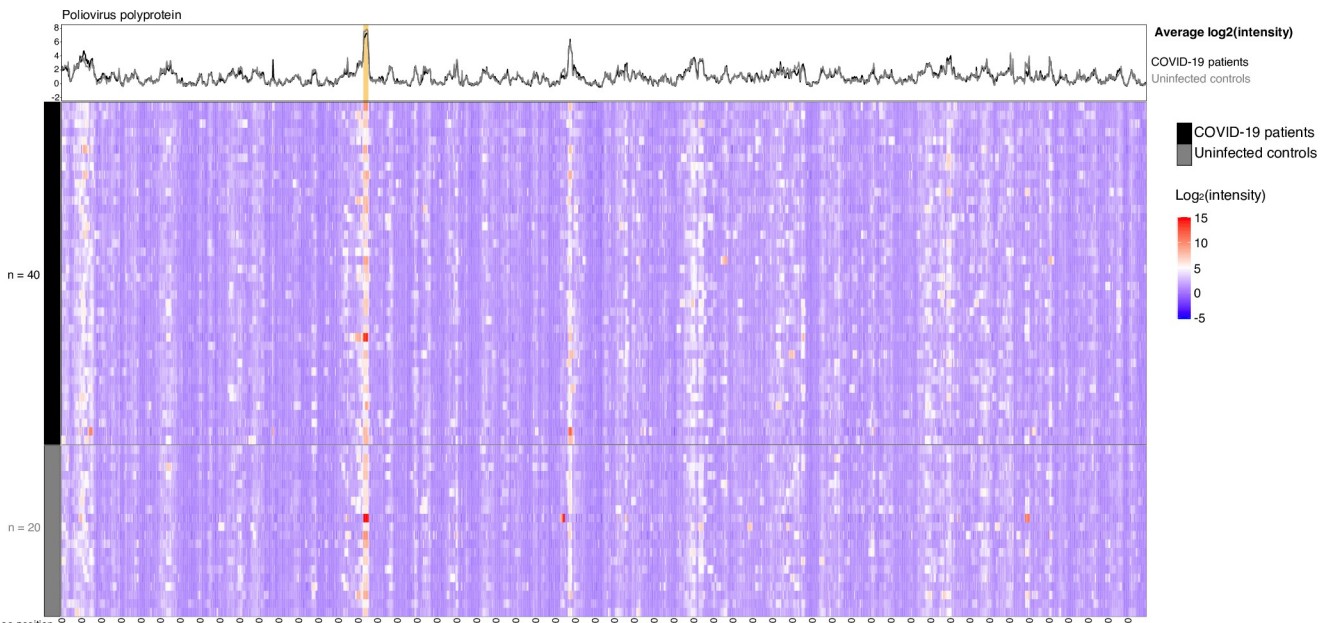

**Fig 1. Patients and controls show reactivity to a poliovirus control.** Sera from 20 controls collected before 2019 were assayed for IgG binding to the full proteome of human poliovirus 1 on a peptide microarray. Binding was measured as reactivity that was >3.00 standard deviations above the mean for the $\log_2$-quantile normalized array data. Patients and controls alike showed reactivity to a well-documented linear poliovirus epitope (start position 613 [IEDB. org]; orange shading in line plot). The data used in this analysis can be accessed online at: https://github.com/Ong-Research/UW_Adult_Covid-19. COVID-19, coronavirus disease 2019; IgG, immunoglobulin G.

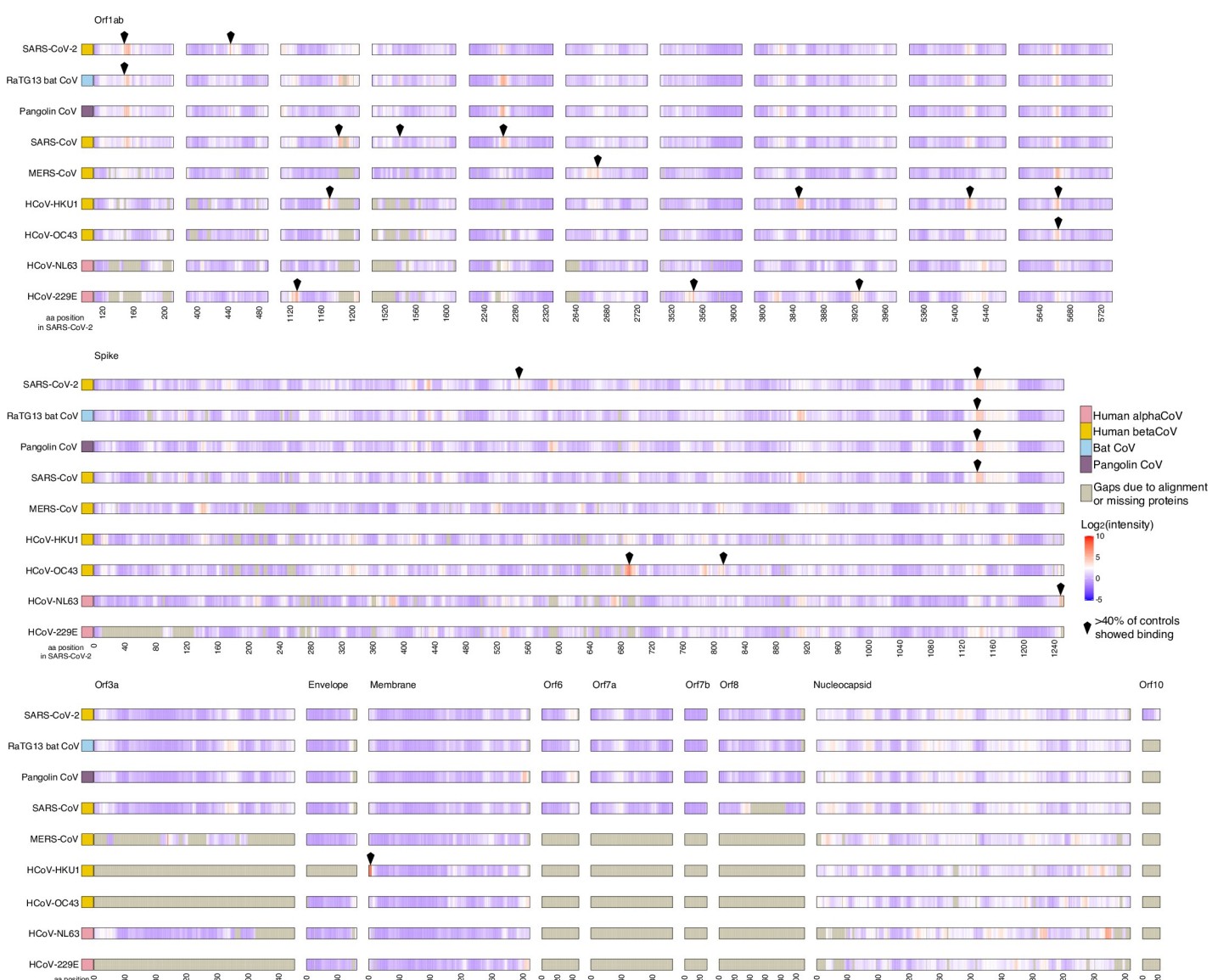

**Fig 2. Control sera show reactivity to CCCoVs and to SARS-CoV, MERS-CoV, and SARS-CoV-2.** Sera from 20 controls collected before 2019 were assayed for IgG binding to the full proteomes of 9 CoVs on a peptide microarray. Viral proteins are shown aligned to the SARS-CoV-2 proteome with each virus having an individual panel; SARS-CoV-2 aa position is represented on the x-axis. Binding was measured as reactivity that was >3.00 standard deviations above the mean for the log₂-quantile normalized array data. Peptides for which >40% of the controls showed binding are indicated by a black diamond. The data used in this analysis can be accessed online at: https://github.com/Ong-Research/UW_Adult_Covid-19. aa, amino acid; CCCoVs, "common cold" CoVs; CoV, coronavirus; IgG, immunoglobulin G; MERS-CoV, Middle Eastern respiratory syndrome coronavirus; SARS-CoV, severe acute respiratory syndrome coronavirus; SARS-CoV-2, severe acute respiratory syndrome coronavirus 2.

previously been shown to be potently neutralizing [32–34], and all 4 of these were ranked within the top 10 of our 79 epitopes. Forty-two of our detected epitopes (including 1-M-24, 553-S-26, 624-S-23, 807-S-26, and 1140-S-25; Table 1) confirm bioinformatic predictions of antigenicity based on SARS-CoV and MERS-CoV [7,8,35–37], including each of the 12 top-ranking epitopes.

The highest specificity (100%) and sensitivity (98%), determined by linear discriminant analysis leave-one-out cross-validation, for any individual peptide was observed for a 16-mer within the 1-M-24 epitope: ITVEELKKLLEQWNLV (S2 Table). Fifteen additional individual

**Table 1. Profiling antibody binding in 40 COVID-19 convalescent patients compared to 20 naïve controls identifies B cell epitopes in SARS-CoV-2 (all data is log$_2$-normalized).**

| Protein | First aa position | Epitope identifier | Sequence | Minimum p-value | Minimum signal | Maximum signal | Mean signal | Mean fold change | Shown to be neutralizing (PMID) | Bioinformatically predicted (PMID) | Pearson correlation of binding with neutralization titer |
|---|---|---|---|---|---|---|---|---|---|---|---|
| M | 1 | 1-M-24 | MADSNGTITVEELKKLLEQWNLVI | 2.19E-23 | 3.45 | 5.97 | 4.45 | 4.77 | | 32183941 | 0.77 |
| N | 384 | 384-N-33 | QRQKKQQTVTLLPAADLDDFSKQLQQSMSSADS | 9.10E-18 | 2.63 | 5.63 | 4.48 | 3.87 | | 32183941 | 0.73 |
| S | 568 | 568-S-26 | DIADTTDAVRDPQTLEILDITPCSFG | 1.61E-14 | 0.64 | 4.44 | 3.55 | 4.83 | | 32843695 | 0.77 |
| S | 1247 | 1247-S-27 | CCSCGSCCKFDEDDSEPVLKGVKLHYT | 7.33E-13 | 2.06 | 4.98 | 3.77 | 3.96 | | 32183941 | 0.69 |
| N | 208 | 208-N-31 | ARMAGNGGDAALALLLDRLNQLESKMSGKG | 1.15E-12 | 0.95 | 3.93 | 2.45 | 2.60 | | 32183941 | 0.76 |
| S | 807 | 807-S-26 | PDPSKPSKRSFIEDLLFNKVTLADAG | 4.68E-12 | 3.75 | 6.29 | 5.33 | 3.62 | 32483236 | 32183941 | 0.69 |
| S | 553 | 553-S-26 | TESNKKFLPFQQFGRDIADTTDAVRD | 7.22E-12 | 2.01 | 5.79 | 4.41 | 3.66 | 32483236; 32612199; 32895485 | 32183941; 32843695 | 0.81 |
| S | 785 | 785-S-27 | VKQIYKTPPIKDFGGFNFSQILPDPSK | 3.42E-11 | 2.22 | 4.95 | 3.80 | 2.99 | | 32183941; 32843695 | 0.67 |
| S | 1140 | 1140-S-25 | PLQPELDSFKEELDKYFKNHTSPDV | 1.08E-09 | 3.40 | 6.71 | 5.84 | 3.16 | 32612199 | 32183941; 32843695 | 0.63 |
| S | 624 | 624-S-23 | IHADQLTPTWRVYSTGSNVFQTR | 8.75E-09 | 0.84 | 2.25 | 1.50 | 1.98 | 32612199 | 32183941 | 0.70 |
| M | 181 | 181-M-32 | LGASQRVAGDSGFAAYSRYRIGNYKLNTDHSS | 1.35E-08 | 1.44 | 3.86 | 2.34 | 2.61 | | 32183941 | 0.66 |
| N | 28 | 28-N-28 | QNGERSGARSKQRRPQGLPNNTASWFTA | 1.78E-08 | 1.75 | 4.13 | 2.94 | 1.86 | | 32183941 | 0.52 |
| ORF1ab | 4514 | 4514-ORF1ab-16 | YTMADLVYALRHFDEG | 3.44E-08 | 2.88 | 2.88 | 2.88 | 3.84 | | | 0.76 |
| M | 152 | 152-M-26 | AGHHLGRCDIKDLPKEITVATSRTLS | 6.33E-08 | 2.96 | 4.26 | 3.44 | 2.96 | | 32183941 | 0.60 |
| S | 549 | 549-S-18 | TGVLTESNKKFLPFQQFG | 1.14E-07 | 4.35 | 4.96 | 4.72 | 3.00 | | 32183941 | 0.73 |
| S | 685 | 685-S-25 | RSVASQSIIAYTMSLGAENSVAYSN | 3.26E-07 | 1.18 | 2.95 | 2.28 | 2.16 | | 32183941 | 0.73 |
| N | 249 | 249-N-18 | KSAAEASKKPRQKRTATK | 2.03E-06 | 2.10 | 3.39 | 2.72 | 1.60 | | 32183941 | 0.51 |
| M | 205 | 205-M-18 | KLNTDHSSSSDNIALLVQ | 2.64E-06 | 2.14 | 3.32 | 2.41 | 2.93 | | 32183941 | 0.65 |
| ORF1ab | 5999 | 5999-ORF1ab-17 | ITREEAIRHVRAWIGF | 4.02E-06 | 1.56 | 1.56 | 1.56 | 2.17 | | | 0.66 |
| ORF1ab | 1239 | 1239-ORF1ab-18 | VTTTLEETKFLTENLLLY | 6.69E-06 | 1.38 | 1.55 | 1.49 | 1.60 | | | 0.45 |
| ORF1ab | 2309 | 2309-ORF1ab-16 | ITISSFKWDLTAFGLV | 6.79E-06 | 1.05 | 1.05 | 1.05 | 2.08 | | | 0.61 |
| S | 613 | 613-S-25 | QDVNCTEVPVAIHADQLTPTWRVYS | 8.65E-06 | 1.52 | 2.90 | 2.35 | 2.12 | | 32183941 | 0.66 |
| ORF1ab | 1551 | 1551-ORF1ab-16 | ITFDNLKTLLSLREVR | 1.46E-05 | 1.01 | 1.01 | 1.01 | 1.75 | | | 0.65 |
| ORF1ab | 6057 | 6057-ORF1ab-17 | DFSRVSAKPPPGDQFKH | 6.02E-05 | 2.85 | 3.32 | 3.09 | 1.42 | | | 0.53 |
| N | 153 | 153-N-26 | NNAAIVLQLPQGTTLPKGFYAEGSRG | 7.56E-05 | 1.91 | 6.02 | 4.70 | 3.17 | | 32183941 | 0.59 |
| ORF1ab | 1720 | 1720-ORF1ab-16 | KTVGELGDVRETMSYL | 9.05E-05 | 1.74 | 1.74 | 1.74 | 1.41 | | | 0.53 |
| S | 635 | 635-S-20 | VYSTGSNVFQTRAGCLIGAE | 9.25E-05 | 0.98 | 1.91 | 1.35 | 1.33 | | 32183941 | 0.53 |
| N | 14 | 14-N-17 | RITFGGPSDSTGSNQNG | 9.46E-05 | 3.35 | 3.82 | 3.59 | 2.16 | | | 0.51 |
| N | 7 | 7-N-21 | QNQRNAPRITFGGPSDSTGSN | 1.70E-04 | 3.24 | 3.70 | 3.39 | 2.34 | | 32183941 | 0.56 |
| S | 940 | 940-S-16 | STASALGKLQDVVNQN | 1.75E-04 | 2.53 | 2.53 | 2.53 | 2.05 | | | 0.57 |
| S | 1155 | 1155-S-20 | YFKNHTSPDVDLGDISGINA | 2.16E-04 | 2.44 | 4.07 | 3.27 | 2.16 | | 32183941 | 0.66 |
| N | 338 | 338-N-19 | KLDDKDPNFKDQVILLNKH | 3.62E-04 | 1.89 | 2.45 | 2.22 | 1.76 | | | 0.56 |
| S | 404 | 404-S-18 | GDEVRQIAPGQTGKIADY | 3.77E-04 | 2.00 | 2.72 | 2.36 | 1.58 | | 32183941; 32843695 | 0.44 |
| ORF8 | 60 | 60-ORF8-20 | LCVDEAGSKSPIQYIDIGNY | 4.24E-04 | 2.31 | 3.18 | 2.66 | 1.82 | | | 0.45 |

*(Continued)*

**Table 1.** (Continued)

| Protein | First aa position | Epitope identifier | Sequence | Minimum p-value | Minimum signal | Maximum signal | Mean signal | Mean fold change | Shown to be neutralizing (PMID) | Bioinformatically predicted (PMID) | Pearson correlation of binding with neutralization titer |
|---|---|---|---|---|---|---|---|---|---|---|---|
| N | 376 | 376-N-22 | ADETQALPQRQKKQQTVTLLPA | 5.98E-04 | 2.31 | 2.92 | 2.67 | 1.93 | | 32183941 | 0.43 |
| ORF3a | 252 | 252-ORF3a-24 | SSGVVNPVMEPIYDEPTTTTSVPL | 7.03E-04 | 3.20 | 4.23 | 3.52 | 2.16 | | | 0.56 |
| N | 230 | 230-N-21 | LESKMSGKGQQQGQTVTKKS | 8.70E-04 | 2.88 | 3.96 | 3.28 | 2.10 | | 32183941 | 0.39 |
| N | 94 | 94-N-16 | IRGGDGKMKDLSPRWY | 1.33E-03 | 2.64 | 2.64 | 2.64 | 1.60 | | 32183941 | 0.61 |
| N | 356 | 356-N-16 | HIDAYKTFPPTEPKKD | 2.06E-03 | 3.30 | 3.30 | 3.30 | 1.84 | | | 0.47 |
| S | 536 | 536-S-17 | NKCVNFNFNGLTGTGVL | 2.10E-03 | 1.81 | 1.95 | 1.88 | 1.15 | | | 0.45 |
| ORF8 | 12 | 12-ORF8-16 | TVAAFHQECSLQSCTQ | 2.24E-03 | 1.03 | 1.03 | 1.03 | 1.00 | | | 0.41 |
| S | 798 | 798-S-17 | GGFNFSQILPDPSKPSK | 2.98E-03 | 2.75 | 3.17 | 2.96 | 1.38 | | 32183941 | 0.39 |
| N | 227 | 227-N-17 | LNQLESKMSGKGQQQQG | 3.53E-03 | 3.39 | 3.51 | 3.45 | 1.94 | | | 0.32 |
| ORF8 | 66 | 66-ORF8-18 | GSKSPIQYIDIGNYTVSC | 3.99E-03 | 2.00 | 2.39 | 2.20 | 1.52 | | | 0.37 |
| ORF1ab | 4451 | 4451-ORF1ab-16 | KDEDDNLIDSYFVVKR | 4.33E-03 | 0.97 | 0.97 | 0.97 | 1.19 | | | 0.32 |
| S | 289 | 289-S-17 | VDCALDPLSETKCTLKS | 5.49E-03 | 2.19 | 2.20 | 2.19 | 1.30 | | 32183941 | 0.60 |
| N | 117 | 117-N-19 | PEAGLPYGANKDGIIWVAT | 5.73E-03 | 0.95 | 2.73 | 1.67 | 1.56 | | | 0.57 |
| M | 175 | 175-M-20 | TLSYYKLGASQRVAGDSGFA | 5.78E-03 | 1.88 | 2.65 | 2.35 | 1.30 | | | 0.39 |
| S | 644 | 644-S-16 | QTRAGCLIGAEHVNNS | 6.88E-03 | 1.66 | 1.66 | 1.66 | 1.22 | | 32183941 | 0.47 |
| ORF6 | 9 | 9-ORF6-16 | VTIAEILLIMRTFKV | 6.98E-03 | 1.07 | 1.07 | 1.07 | 1.09 | | | 0.54 |
| N | 242 | 242-N-19 | QGGTVTKKSAAEASKKPRQ | 7.50E-03 | 2.98 | 3.15 | 3.09 | 1.42 | | 32183941 | 0.46 |
| S | 656 | 656-S-17 | VNNSYECDIPIGAGICA | 8.91E-03 | 3.32 | 3.36 | 3.34 | 1.79 | | 32183941 | 0.62 |
| S | 541 | 541-S-16 | FNFNGLTGTGVLTESN | 9.09E-03 | 1.68 | 1.68 | 1.68 | 1.36 | | | 0.46 |
| S | 844 | 844-S-16 | IAARDLICAQKFNGLT | 9.63E-03 | 1.93 | 1.93 | 1.93 | 1.19 | | 32183941 | 0.64 |
| S | 804 | 804-S-17 | QILPDPSKPSKRSFIED | 1.42E-02 | 2.61 | 2.97 | 2.79 | 1.45 | | | 0.28 |
| N | 126 | 126-N-17 | NKDGIIWVATEGALNTP | 1.43E-02 | 1.16 | 1.40 | 1.28 | 1.11 | | | 0.43 |
| N | 96 | 96-N-17 | GGDGKMKDLSPRWYFYY | 1.49E-02 | 0.85 | 1.41 | 1.13 | 1.19 | | 32183941 | 0.46 |
| ORF3a | 235 | 235-ORF3a-17 | KIVDEPEEHVQIHTIDG | 1.56E-02 | 1.20 | 1.20 | 1.20 | 1.46 | | | 0.27 |
| N | 122 | 122-N-16 | PYGANKDGIIWVATEG | 1.95E-02 | 0.85 | 0.85 | 0.85 | 1.12 | | | 0.54 |
| ORF8 | 53 | 53-ORF8-16 | KSAPLIELCVDEAGSK | 1.95E-02 | 1.43 | 1.43 | 1.43 | 1.03 | | | 0.08 |
| N | 124 | 124-N-16 | GANKDGIIWVATEGAL | 2.55E-02 | 0.91 | 0.91 | 0.91 | 1.03 | | | 0.47 |
| N | 336 | 336-N-16 | AIKLDDKDPNFKDQVI | 2.58E-02 | 3.31 | 3.31 | 3.31 | 1.43 | | | 0.43 |
| ORF1ab | 1546 | 1546-ORF1ab-16 | LDGEVITFDNLKTLLS | 3.75E-02 | 0.94 | 0.94 | 0.94 | 1.07 | | | 0.20 |
| S | 306 | 306-S-16 | FTVEKGIYQTSNFRVQ | 4.00E-02 | 1.92 | 1.92 | 1.92 | 1.38 | | 32183941 | 0.46 |
| S | 241 | 241-S-16 | LLALHRSYLTPGDSSS | 4.06E-02 | 1.12 | 1.12 | 1.12 | 1.06 | | 32183941 | 0.47 |
| S | 768 | 768-S-18 | TGIAVEQDKNTQEVFAQV | 4.33E-02 | 3.23 | 4.15 | 3.76 | 2.16 | | 32183941 | 0.25 |
| ORF3a | 16 | 16-ORF3a-16 | KQGEIKDATPSDFVRA | 4.40E-02 | 3.40 | 3.40 | 3.40 | 1.87 | | | 0.43 |
| S | 1164 | 1164-S-16 | VDLGDISGINASVVNI | 5.04E-02 | 3.29 | 3.29 | 3.29 | 1.55 | | 32183941 | 0.53 |
| S | 172 | 172-S-16 | SQPFLMDLEGKQGNFK | 5.10E-02 | 2.69 | 2.69 | 2.69 | 1.09 | | 32183941 | 0.37 |
| ORF3a | 21 | 21-ORF3a-16 | KDATPSDFVRATATIP | 5.25E-02 | 2.03 | 2.03 | 2.03 | 1.43 | | | 0.40 |
| ORF1ab | 2584 | 2584-ORF1ab-16 | AEVAVKMFDAYVNTFS | 6.57E-02 | 1.54 | 1.54 | 1.54 | 1.04 | | | 0.20 |
| S | 1178 | 1178-S-16 | NIQKEIDRLNEVAKNL | 7.05E-02 | 4.19 | 4.19 | 4.19 | 1.63 | | | 0.60 |
| S | 661 | 661-S-16 | ECDIPIGAGICASYQT | 7.17E-02 | 2.35 | 2.35 | 2.35 | 1.32 | | 32183941 | 0.56 |

(*Continued*)

**Table 1.** (Continued)

| Protein | First aa position | Epitope identifier | Sequence | Minimum p-value | Minimum signal | Maximum signal | Mean signal | Mean fold change | Shown to be neutralizing (PMID) | Bioinformatically predicted (PMID) | Pearson correlation of binding with neutralization titer |
|---|---|---|---|---|---|---|---|---|---|---|---|
| ORF3a | 18 | 18-ORF3a-16 | GEIKDATPSDFVRATA | 7.68E-02 | 2.57 | 2.57 | 2.57 | 1.66 | | | 0.33 |
| S | 410 | 410-S-16 | IAPGQTGKIADYNYKL | 7.72E-02 | 3.23 | 3.23 | 3.23 | 1.35 | | 32183941; 32843695 | 0.33 |
| S | 1161 | 1161-S-17 | SPDVDLGDISGINASVV | 7.84E-02 | 3.88 | 4.49 | 4.18 | 1.75 | | 32183941 | 0.50 |
| S | 761 | 761-S-16 | TQLNRALTGIAVEQDK | 8.12E-02 | 3.34 | 3.34 | 3.34 | 2.10 | | 32183941 | 0.24 |
| ORF1ab | 1681 | 1681-ORF1ab-16 | LTLQQIELKFNPPALQ | 8.28E-02 | 1.64 | 1.64 | 1.64 | 1.19 | | | 0.56 |
| ORF1ab | 1572 | 1572-ORF1ab-16 | TTVDNINLHTQVVDMS | 8.60E-02 | 2.28 | 2.28 | 2.28 | 1.40 | | | 0.32 |

aa, amino acid; COVID-19, coronavirus disease 2019; M, membrane; N, nucleocapsid; S, spike; SARS-CoV-2, severe acute respiratory syndrome coronavirus 2.

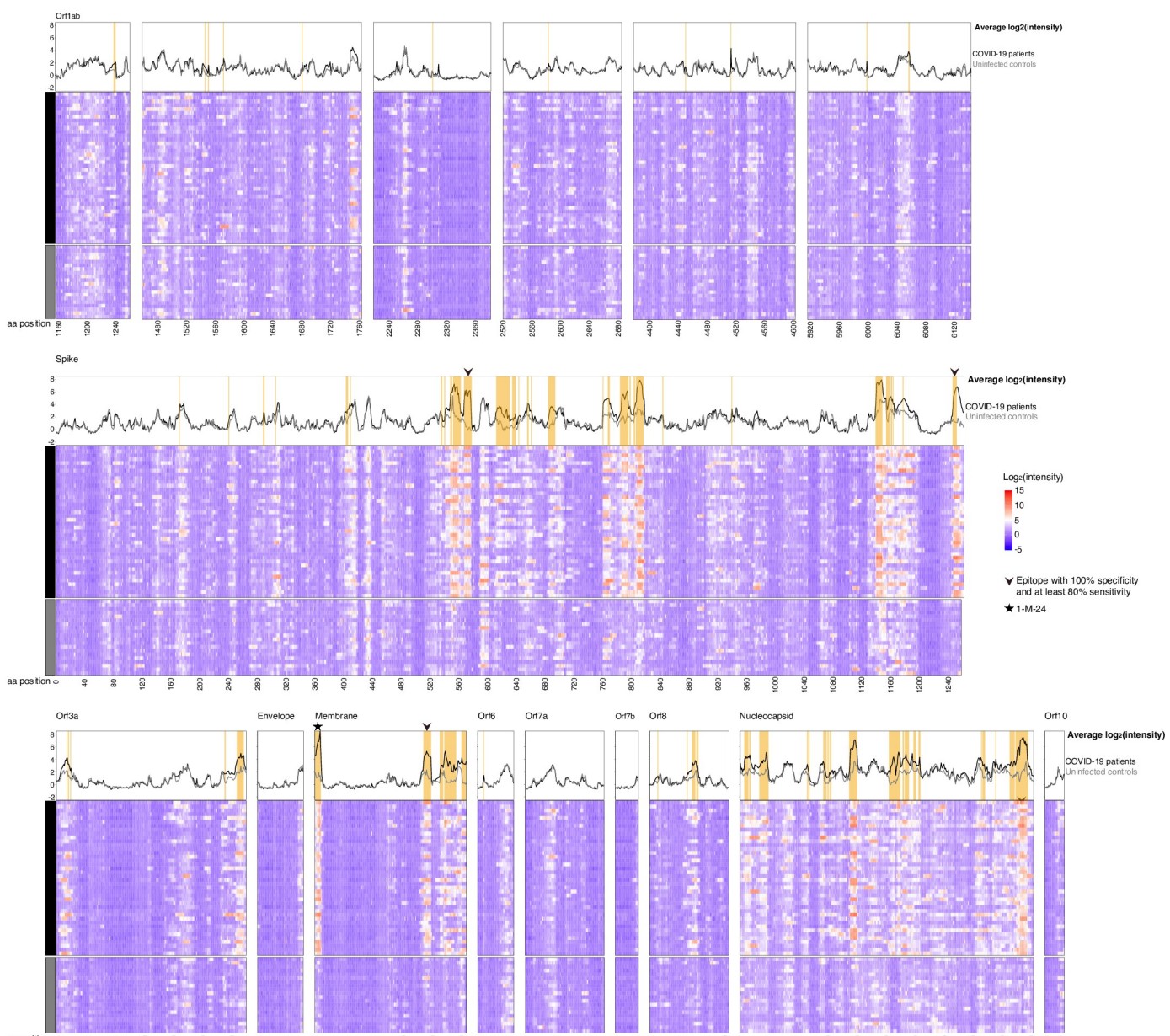

**Fig 3. Anti-SARS-CoV-2 antibodies bind throughout the viral proteome.** Sera from 40 COVID-19 convalescent patients were assayed for IgG binding to the full SARS-CoV-2 proteome on a peptide microarray. B cell epitopes were defined as peptides in which patients' average $\log_2$-normalized intensity (black lines in line plots) is 2-fold greater than controls' (gray lines in line plots), and $t$ test statistics yield adjusted $p$-values <0.1; epitopes are identified by orange shading in the line plots. Epitopes having at least 100% specificity and at least 80% sensitivity for SARS-CoV-2 infection are indicated by a black arrow. The 1-M-24 epitope, which had the highest combined reactivity, specificity, and sensitivity of all epitopes we defined, is indicated by a black star. The data used in this analysis can be accessed online at: https://github.com/Ong-Research/UW_Adult_Covid-19. aa, amino acid; COVID-19, coronavirus disease 2019; IgG, immunoglobulin G; SARS-CoV-2, severe acute respiratory syndrome coronavirus 2.

peptides in M, S, and N had 100% measured specificity and at least 80% sensitivity (Table 2). Combinations of 1-M-24 with 1 of 5 other epitopes (384-N-33, 807-S-26, 6057-ORF1ab-17, 227-N-17, and 4451-ORF1ab-16) yielded an area under the curve receiver operating characteristic of 1.00 (S3 Table) based on linear discriminant analysis leave-one-out-cross-validation.

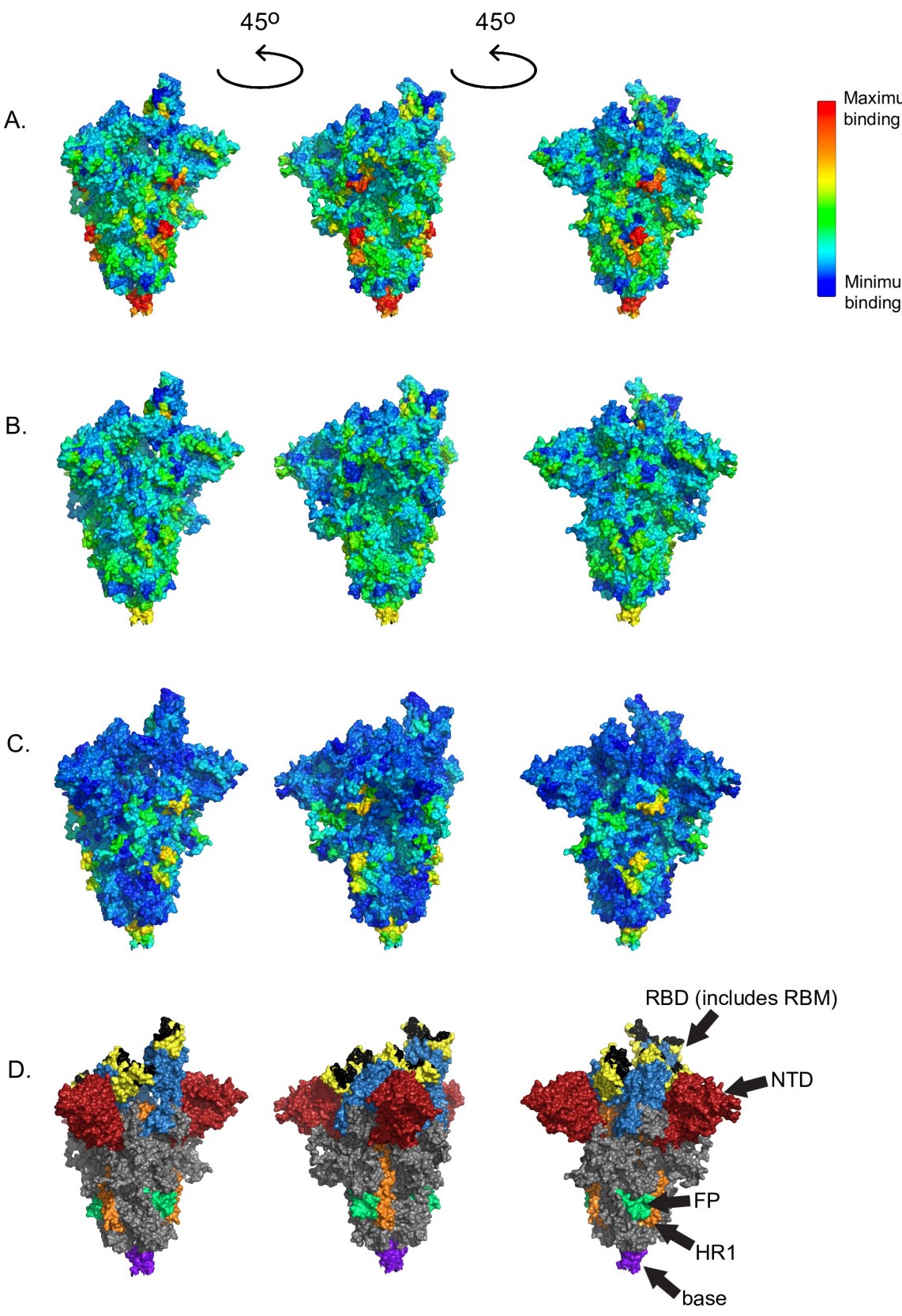

**Fig 4. Anti-SARS-CoV-2 antibodies to S protein show the highest binding in the fusion cleavage site.** Binding reactivities were localized on a coordinate file for a trimer of the SARS-CoV-2 S protein using a dark blue (low, 0.00 fluorescence intensity) to red (high, 9.00 fluorescence intensity) color scale. (A) COVID-19 convalescent patients, (B) naïve controls, and (C) the difference between patients and controls are shown. The highest reactivity occurred in the fusion peptide (aa 788–806) and at the base of the extracellular portion of the molecule (aa 984–1163), with lower reactivity in the receptor-binding domain (aa 319–541). (D) Key regions of the S protein are labeled and colored. In the S1 subunit (aa 14–685), the NTD (aa 14–305) is red, and the RBD (aa 319–541) is blue. Within the RBD, the RBM (aa 438–506) is yellow, and the residues that bind to the ACE2 receptor (aa 446, 449, 453, 455–456, 473, 475–476, 484, 486–487, 489–490, 493, 496, 498, 500–502, and 505) are in black. In the S2 subunit (aa 686–1273), the FP (aa 788–806) is green, the HR1 (aa 912–984) is orange, and the base of the extracellular part of the protein (base, roughly aa 1140–1160) is purple [6,38]. The remainder of the protein is gray. The data used in this analysis can be accessed online at: https://github.com/Ong-Research/UW_Adult_Covid-19. AA, amino acid; ACE2, angiotensin converting enzyme 2; COVID-19, coronavirus disease 2019; FP, fusion peptide; HR1, heptad repeat 1; NTD, N-terminal domain; RBD, receptor-binding domain; RBM, receptor-binding motif; S, spike; SARS-CoV-2, severe acute respiratory syndrome coronavirus 2.

## Anti-SARS-CoV-2 antibodies may cross-reactively bind peptides in other CoVs

We determined epitopes bound by anti-SARS-CoV-2 antibodies in non-SARS-CoV-2 CoVs by the same criteria we used to determine epitopes in SARS-CoV-2. Epitopes in these viruses were defined as binding by antibodies in COVID-19 convalescent sera to peptides at an average $\log_2$-normalized intensity at least 2-fold greater than in controls with $t$ test statistics yielding adjusted $p$-values <0.1. Some of these epitopes were identical sequences with SARS-CoV-2, particularly in the RaTG13 bat betacoronavirus (β-CoV), the closest known relative of SARS-CoV-2 (96% nucleotide identity) [39,40], the pangolin CoV (85% nucleotide identity with SARS-CoV-2) [41], and SARS-CoV (78% identity) [39]. Cross-reactivity of an antibody is typically determined by evaluating a pure preparation of specific antibodies or by competition assays. However, since our Wisconsin controls are almost certainly naïve to MERS-CoV, SARS-CoV, and bat and pangolin CoVs, we can make predictions about cross-reactivity (as opposed to binding due to sequence identity).

**Table 2. Sixteen peptides in the SARS-CoV-2 proteome had 100% specificity and at least 80% sensitivity for SARS-CoV-2 infection in 40 COVID-19 convalescent patients compared to 20 naïve controls.**

| Protein | First aa position | Sequence | Specificity | Sensitivity | F1 |
|---------|-------------------|----------|-------------|-------------|-----|
| M | 8 | ITVEELKKLLEQWNLV | 1 | 0.98 | 0.99 |
| M | 7 | TITVEELKKLLEQWNL | 1 | 0.95 | 0.97 |
| N | 390 | QTVTLLPAADLDDFSK | 1 | 0.95 | 0.97 |
| N | 388 | KQQTVTLLPAADLDDF | 1 | 0.90 | 0.95 |
| N | 391 | TVTLLPAADLDDFSKQ | 1 | 0.90 | 0.95 |
| S | 570 | ADTTDAVRDPQTLEIL | 1 | 0.88 | 0.93 |
| S | 571 | DTTDAVRDPQTLEILD | 1 | 0.88 | 0.93 |
| S | 574 | DAVRDPQTLEILDITP | 1 | 0.85 | 0.92 |
| S | 576 | VRDPQTLEILDITPCS | 1 | 0.85 | 0.92 |
| S | 1253 | CCKFDEDDSEPVLKGV | 1 | 0.85 | 0.92 |
| S | 572 | TTDAVRDPQTLEILDI | 1 | 0.83 | 0.90 |
| S | 573 | TDAVRDPQTLEILDIT | 1 | 0.83 | 0.90 |
| S | 577 | RDPQTLEILDITPCSF | 1 | 0.83 | 0.90 |
| S | 1252 | SCCKFDEDDSEPVLKG | 1 | 0.83 | 0.90 |
| M | 162 | KDLPKEITVATSRTLS | 1 | 0.83 | 0.90 |
| S | 1250 | CGSCCKFDEDDSEPVL | 1 | 0.80 | 0.89 |

aa, amino acid; COVID-19, coronavirus disease 2019; M, membrane; N, nucleocapsid; S, spike; SARS-CoV-2, severe acute respiratory syndrome coronavirus 2.

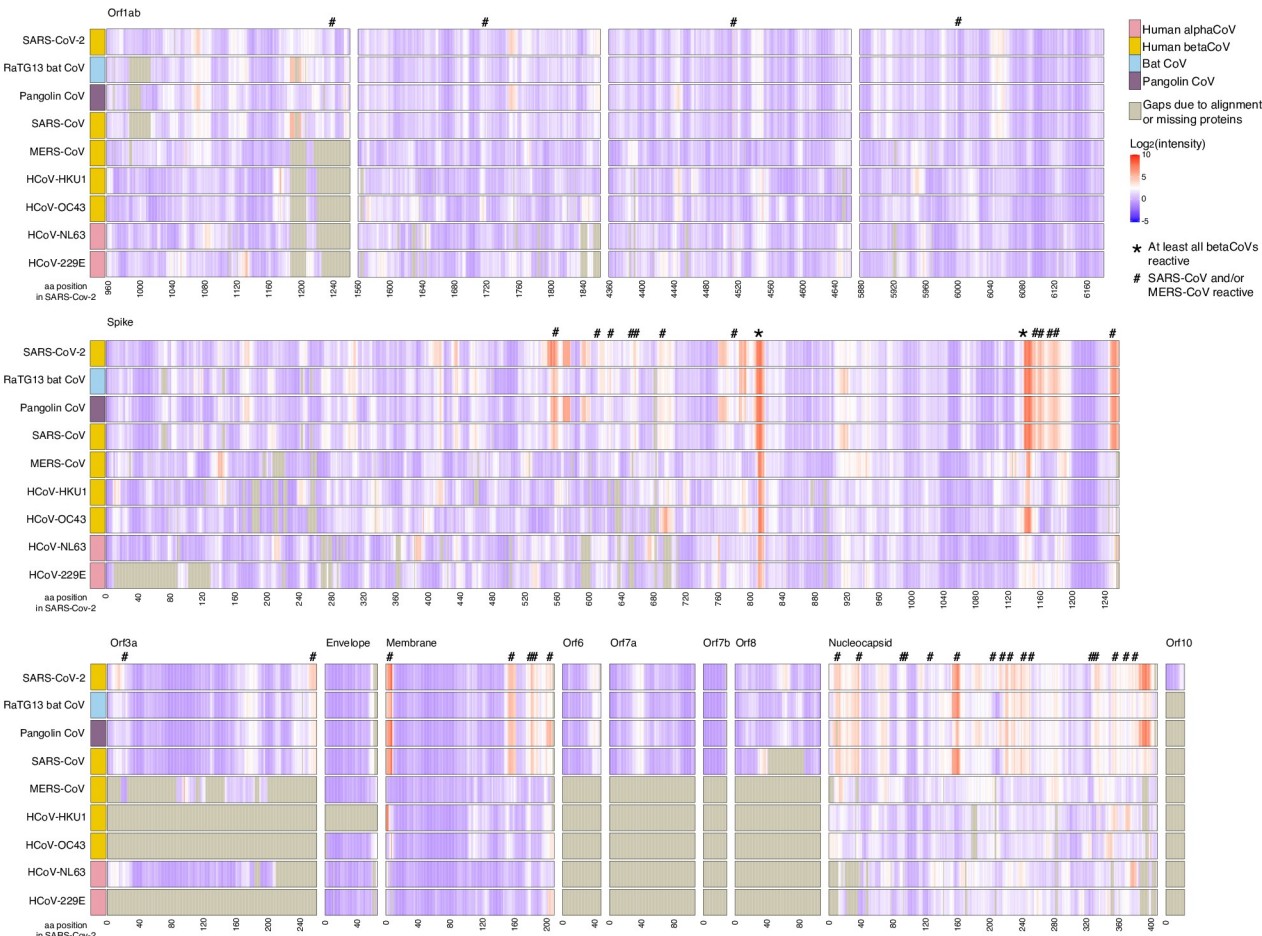

**Fig 5. Anti-SARS-CoV-2 antibodies may cross-react with other CoVs.** Sera from 40 COVID-19 convalescent patients were assayed for IgG binding to 9 CoVs on a peptide microarray; averages for all 40 are shown. Viral proteins are aligned to the SARS-CoV-2 proteome; SARS-CoV-2 aa position is represented on the x-axis. Regions that may be cross-reactive across all β-CoVs (*) or cross-reactive for SARS-CoV or MERS-CoV (#) are indicated. Gray shading indicates gaps due to alignment or lacking homologous proteins. Cross-reactive binding is defined as peptides in which patients' average log₂-normalized intensity is 2-fold greater than controls' and *t* test statistics yield adjusted *p*-values <0.1. The data used in this analysis can be accessed online at: https://github.com/Ong-Research/UW_Adult_Covid-19. aa, amino acid; β-CoV, betacoronavirus; CoV, coronavirus; COVID-19, coronavirus disease 2019; IgG, immunoglobulin G; MERS-CoV, Middle Eastern respiratory syndrome coronavirus; SARS-CoV, severe acute respiratory syndrome coronavirus; SARS-CoV-2, severe acute respiratory syndrome coronavirus 2.

Antibodies in COVID-19-convalescent sera appeared to be cross-reactive with identical or homologous epitopes in S, M, N, ORF1ab, ORF3, ORF6, and ORF8 in other CoVs (Fig 5, S2 Fig, and S3 and S4 Data). Overall, the greatest number of epitopes in any non-SARS-CoV-2 CoV occurred in the RaTG13 bat β-CoV at 74 epitopes (60 identical to SARS-CoV-2, 13 homologous nonidentical, 1 without a homologous SARS-CoV-2 epitope). The second greatest number, 60 epitopes, occurred in the pangolin CoV (23 identical to SARS-CoV-2, 30 homologous nonidentical, 6 without a homologous SARS-CoV-2 epitope, 1 without a homologous region in SARS-CoV-2), and third SARS-CoV with 45 epitopes, (10 identical to SARS-CoV-2, 32 homologous nonidentical, 3 without a homologous SARS-CoV-2 epitope) (S3 and S4 Data). Most (8 of 12) of the epitopes that were not in areas having epitopes in the homologous SARS-CoV-2 region occurred in ORF1ab, with the others occurring in S (2 epitopes) and N (2 epitopes). These epitopes were not conserved among each other and were not conserved with any epitopes in the CCCoVs (S3 Data).

One region, corresponding to SARS-CoV-2 epitope 807-S-26, showed binding or potential cross-reactivity across all CoVs, and one, corresponding to SARS-CoV-2 epitope 1140-S-25, showed binding or potential cross-reactivity across all β-CoVs (Fig 5). Epitope 807-S-26 includes the CoV S fusion peptide, and 1140-S-25 is immediately adjacent to the heptad repeat region 2, both of which are involved in membrane fusion [42].

## Enzyme-linked immunosorbent assays (ELISAs) confirm peptide microarray findings

Having determined reactivity and apparent cross-reactivity by peptide array, we aimed to independently confirm and validate these findings by ELISA. We selected 4 peptides for ELISA evaluation (1253-S-16, 814-S-16, 8-M-16, and 390-N-16) from those in our top 10 ranked epitopes, considering diversity among the proteins represented, association with neutralizing capacity, and potential cross-reactivity across multiple CoVs, and using the 16-mer in each epitope that most correctly discriminated between patients and controls. All 4 SARS-CoV-2 peptides had higher IgG binding in COVID-19 convalescent sera than in controls (Fig 6). Peptide 8-M-16 showed the greatest discrimination between COVID-19 convalescent and control sera with only 3 COVID-19 convalescent samples having values similar to controls. Both peptides 1253-S-16 and 814-S-16 showed greater binding in controls than either 8-M-16 or 390-N-16, confirming our findings of greater potential cross-reactivity among epitopes found in S.

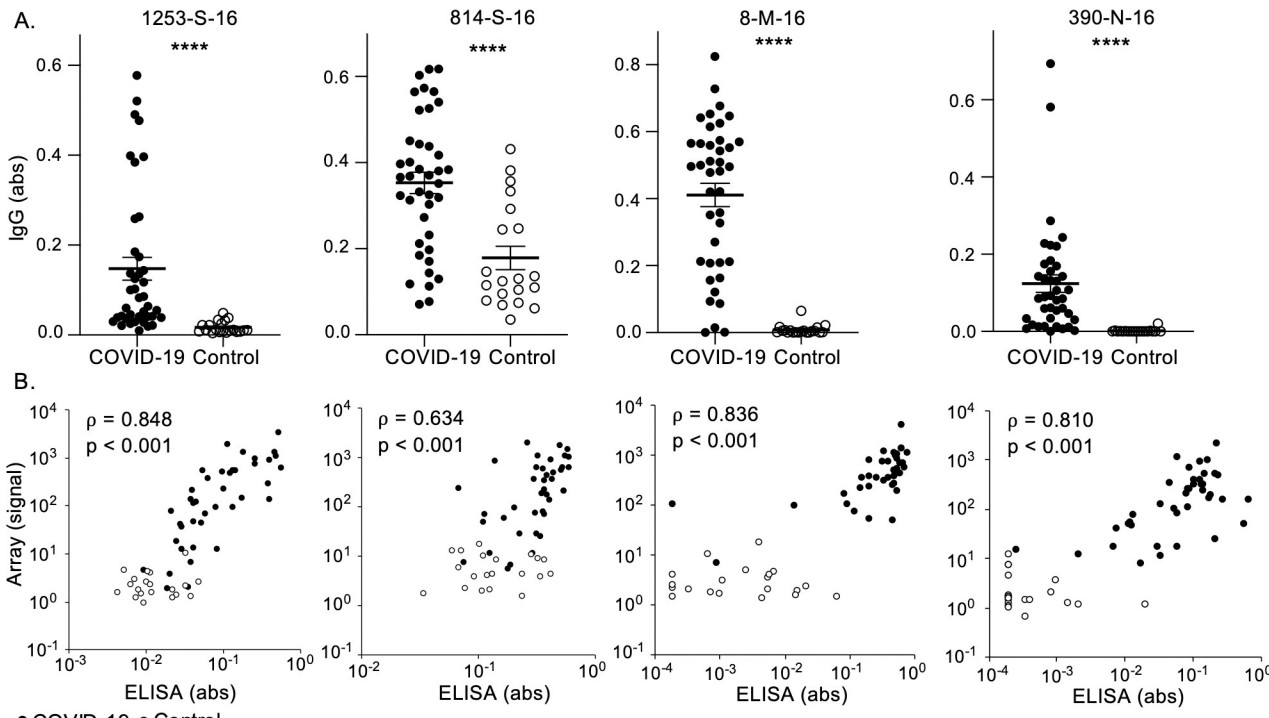

**Fig 6. Higher IgG binding to SARS-CoV-2 peptides in COVID-19 convalescent patients compared to controls by ELISA.** (A) IgG binding to SARS-CoV-2 peptides in COVID-19 convalescent ($n = 40$) and naïve control ($n = 20$) sera was measured by ELISA. Bars indicate mean abs +/− SEM and ****$p < 0.0001$ by $t$ test. (B) Anti-SARS-CoV-2 peptide IgG detected by ELISA was compared to array findings by Spearman rank-order correlation (Spearman correlation coefficient, ρ) for COVID-19 convalescent ($n = 40$, closed circles) and control ($n = 20$, open circles) sera. The data used in this analysis can be accessed online at: https://github.com/Ong-Research/UW_Adult_Covid-19. abs, absorbance; COVID-19, coronavirus disease 2019; ELISA, enzyme-linked immunosorbent assay; IgG, immunoglobulin G; SARS-CoV-2, severe acute respiratory syndrome coronavirus 2; SEM, standard error of the mean.

### Reactivity in some epitopes correlates with disease severity

Increased antibody titer and duration have been associated with increased severity of illness due to infection with SARS-CoV-2 [43–47] and other CoVs [48], although data on epitope-level differences by severity is lacking [49]. We compared reactivity in patients within our cohort whose COVID-19 course required intubation and mechanical ventilation ($n = 8$) with reactivity in COVID-19 convalescent patients who never required hospitalization ($n = 25$) using multilinear regression accounting for age, sex, immunocompromising conditions, and Charlson comorbidity index score [50] to determine epitope-level resolution of differences in reactivity. Nine epitopes in S (2 epitopes), M (1 epitope), N (2 epitopes), and ORF3a (4 epitopes) showed statistically significant ($p < 0.05$) increases in reactivity for intubated patients relative to never-hospitalized patients (Fig 7, S4 Table). The S epitopes (289-S-17 and 613-S-25) both occurred in the S1 subunit (aa 14–685), with one (289-S-17) in the N-terminal domain [6] (see Fig 4D), whose function is not well understood but which may play a role in membrane fusion [51]. The M epitope (1-M-24) was the highly reactive epitope in the N-terminus of this protein discussed above. The N epitopes (336-N-16 and 376-N-22) occurred in the C-terminal domain (336-N-16), which is thought to bind nucleic acids, and in the unstructured C-tail (376-N-22) [52]. The ORF3a epitopes clustered near the N-terminus of the protein (16-ORF3a-16, 18-ORF3a-16, and 21-ORF3a-16) with one other epitope nearer the C-terminus (252-ORF3a-24). No epitopes showed statistically significant increases in reactivity for never-hospitalized patients relative to intubated patients (S4 Table).

## Discussion

In our analysis of antibody binding to the full proteome of SARS-CoV-2, the highest magnitude binding of anti-SARS-CoV-2 antibodies from human sera occurred for an epitope in the

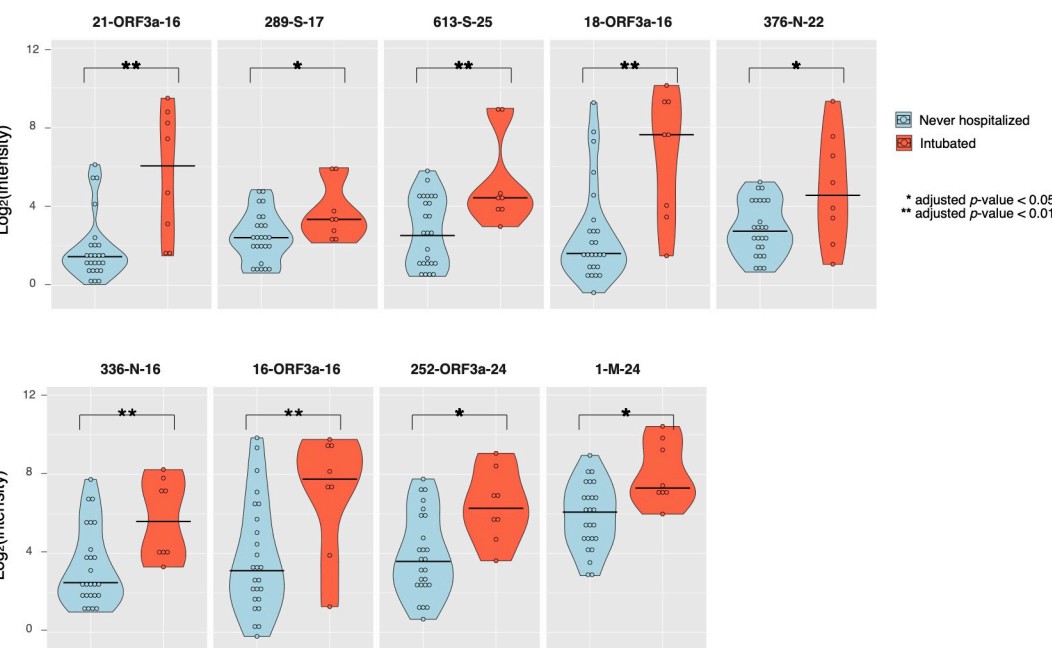

**Fig 7. Disease severity correlates with increased antibody binding in specific SARS-CoV-2 epitopes.** IgG reactivity against SARS-CoV-2 epitopes identified by peptide microarray in COVID-19 convalescent patients who were never hospitalized versus intubated patients showed statistically significant increases in reactivity in intubated patients for 11 epitopes. The data used in this analysis can be accessed online at: https://github.com/Ong-Research/UW_Adult_Covid-19. COVID-19, coronavirus disease 2019; IgG, immunoglobulin G; SARS-CoV-2, severe acute respiratory syndrome coronavirus 2.

N-terminus of M protein, with high specificity and sensitivity. Antibodies produced after infection with SARS-CoV-2 reacted with epitopes throughout the proteomes of other human and nonhuman CoVs, recognizing homologous regions across all CoVs. Taken together, these results confirm that humans mount strong, broad antibody responses to SARS-CoV-2 proteins in addition to S and N, and they implicate M epitopes as highly relevant to diagnostic and potentially to vaccine design.

M proteins are the most abundant proteins in CoV virions [17]. The N-terminus of M is known in other CoVs to be a small, glycosylated ectodomain that protrudes outside the virion and interacts with S, N, and E [17], while the rest of M resides within the viral particle. Full-length SARS-CoV M has been shown to induce protective antibodies [20,53], and patterns of antibodies binding to SARS-CoV M are similar to those we found in SARS-CoV-2 [35]. SARS-CoV anti-M antibodies can synergize with anti-S and anti-N antibodies for improved neutralization [20,53], and M has been used in protective SARS-CoV and MERS-CoV vaccines [8]. However, the mechanism of protection of anti-M antibodies remains unknown, and this protein remains largely understudied and underutilized as an antigen. Other groups have not previously identified the high magnitude binding we observed for M, though that may be due to other studies' use of samples collected earlier in the course of infection or different techniques, populations, or computational algorithms [54,55]. Notably, some of the highest binding we observed in the S protein occurred at the base of the extracellular portion of the protein, which would be the site of the putative interaction between SARS-CoV-2 S and M. The ACE2 binding site and the RBD in general are not as reactive, by these methods, as expected, suggesting that other, less-investigated epitopes may be playing a larger role in immunity to SARS-CoV-2 than is currently appreciated, which is further bolstered by the correlation of some of this binding with neutralizing titers. Our results, in concert with prior knowledge of anti-SARS-CoV antibodies, strongly suggest that epitopes in M, particularly the 1-M-24 epitope as well as other novel epitopes we identified, should be investigated further as potential targets in SARS-CoV-2 diagnostics, vaccines, and therapeutics.

Among the accessory proteins against which we detected antibodies, ORF8 has been the best studied. The *ORF8* gene is part of a hypervariable region, having undergone multiple substitutions and deletions and being recognized as a recombination hotspot [56]. ORF8 protein is considered to have immunomodulatory activity and has been shown to potently down-regulate major histocompatibility complex class I expression in several cell lines [57] and to antagonize interferon signaling [58,59]. A deletion in ORF8 appears to be associated with a milder clinical COVID-19 course [60], and ORF8 has been shown to be secreted [61], indicating that the epitopes we defined here may merit further investigation for development of potential vaccines and therapeutics. Less is known about the other SARS-CoV-2 accessory proteins in which we found epitopes, ORF3a and ORF6, although some studies have implicated them in immunomodulatory functions [59,62–64]. Further investigations will be needed to determine the function of these proteins, which will provide greater insight into how the epitopes we found may be useful in the development of countermeasures against COVID-19.

Interestingly, we found antibodies bind or bind adjacent to a number of the mutations in some of the "variants of concern" (VOCs) of SARS-CoV-2, which have recently emerged [65–67], so named because they appear to potentially be more transmissible than previous known variants or to escape antibody binding [68]. The epitopes we defined contained or were immediately adjacent to the locations of the majority of the variant-defining mutation sites in the structural and accessory proteins of the B.1.1.7 and B.1.351 variants and to one-quarter of the structural or accessory protein mutations sites of the P.1, B.1.427, and B.1.429 variants (see

S5 Table). These results suggest that antigen escape may be driving the rise and dominance of variants. Recent works have demonstrated this phenomenon with mutations in S [69,70] but have not investigated this possibility for other proteins. Our findings suggest that antibodies to non-S proteins may be important to this process, as well. Given that the VOCs were identified after we performed our experiments, we did not include mutations characteristic of the VOCs in our array design for this current work. Future studies should include representation of the VOCs in order to discern any differences in antibody binding.

We also found that antibodies produced in response to SARS-CoV-2 infection appear to bind peptides representing homologous epitopes throughout the proteomes of other human and nonhuman CoVs. Hundreds of CoVs have been discovered in bats and other species [27,39–41,71,72], making future spillovers inevitable. The potential broad cross-reactivity we observed in some homologous peptide sequences may help guide the development of pan-CoV vaccines [15], especially given that antibodies binding to 807-S-26 and 1140-S-25, which showed potential cross-reactivity across all CoVs and all β-CoVs, respectively, are known to be potently neutralizing [32,33]. A caveat is that our methods cannot discern whether the increased IgG binding to CCCoVs in COVID-19 convalescent sera is due to newly developed cross-reactive antibodies or due to the stimulation of a memory response against the original CCCoV antigens. However, cross-reactivity of anti-SARS-CoV-2 antibodies with SARS-CoV or MERS-CoV is likely real, since our population was very unlikely to have been exposed to those viruses. A more stringent assessment of cross-reactivity as well as functional investigations into these cross-reactive antibodies will be vital in determining their capacity for cross-protection. Further, our methods efficiently detect antibody binding to linear epitopes [73], but their sensitivity for detecting parts of conformational epitopes, which are considered highly important in the immune response to SARS-CoV-2 and which are believed to be the type of epitope found within the RBD [23,25,74–76], is unknown. Additional analyses will be required to determine whether epitopes newly identified here induce neutralizing or otherwise protective antibodies. It is interesting to note that SARS-CoV-2 infection resulted in some antibodies that bound epitopes in other coronaviruses without binding the homologous part of SARS-CoV-2. These epitopes were not conserved with each other (S3 Data), and most of this binding occurred in nonstructural proteins in ORF1ab, which may indicate that this was nonspecific binding resulting from a generalized immune activation.

Finally, we demonstrated that more severely ill patients have significantly greater reactivity to certain epitopes in S, M, N, and ORF3a. The 9 epitopes with significantly higher magnitude reactivity in intubated patients may play a role in the overaggressive immune response known to characterize severe COVID-19 [7,77], suggesting that they may be targets for treatment in or prevention of severe disease. Our data collection included date of first positive test (S1 Table) but not of symptom onset, but future studies that include these data could investigate potential correlations between symptoms and antibody kinetics. Alternatively, the antibody response in general may be higher in very sick patients, expanding the repertoire of antibody reactivity. Future studies should investigate whether these differences can be detected early in the disease course to determine their potential utility as predictive markers of disease severity. Future studies may also investigate these epitopes' potential as targets for medical countermeasures [51], although consideration should be given to the small sample size of our investigation.

Many questions remain regarding the biology and immunology related to SARS-CoV-2. Our extensive profiling of epitope-level resolution antibody reactivity in COVID-19 convalescent patients, confirmed by independent assays, provides new epitopes that could serve as important targets in the development of improved diagnostics, vaccines, and therapeutics against SARS-CoV-2 and dangerous human CoVs that may emerge in the future.

## Methods

### Peptide microarray design and synthesis

Viral protein sequences were selected and submitted to Nimble Therapeutics (Madison, Wisconsin, USA) for development into a peptide microarray [73]. Sequences represented include proteomes of all 7 coronaviruses known to infect humans, proteomes of closely related coronaviruses found in bats and pangolins, and spike proteins from other coronaviruses (accession numbers and replicates per peptide shown in S5 Data). A number of proteins were included as controls, including poliovirus, 7 strains of human rhinovirus, and human cytomegalovirus 65kDa phosphoprotein. We chose these controls given that we expect most human adults will have antibody reactivity to at least one of these proteins and proteomes. Accession numbers used to represent each viral protein are listed in the Supporting information (accession numbers and replicates per peptide shown in S5 Data). All proteins were tiled as 16 amino acid peptides overlapping by 15 amino acids. All unique peptides were tiled in a lawn of thousands of copies, with each unique peptide represented in at least 3 and up to 5 replicates (S5 Data). The peptide sequences were synthesized in situ with a Nimble Therapeutics Maskless Array Synthesizer (MAS) by light-directed solid-phase peptide synthesis using an amino-functionalized support (Geiner Bio-One) coupled with a 6-aminohexanoic acid linker and amino acid derivatives carrying a photosensitive 2-(2-nitrophenyl) propyloxycarbonyl (NPPOC) protection group (Orgentis Chemicals). Unique peptides were synthesized in random positions on the array to minimize impact of positional bias. Each array consists of 12 subarrays, where each subarray can process 1 sample, and each subarray contains up to 389,000 unique peptide sequences.

### Human patients and controls

The study was conducted in accordance with the Declaration of Helsinki and approved by the Institutional Review Board of the University of Wisconsin-Madison. Clinical data and sera from patients infected with SARS-CoV-2 were obtained from the University of Wisconsin (UW) COVID-19 Convalescent Biobank and from controls (sera collected prior to 2019) from the UW Rheumatology Biobank [78]. All patients and controls were 18 years of age or older at the time of recruitment and provided informed consent. COVID-19 convalescent patients had a positive SARS-COV-2 PCR test at UW Health with sera collected 5 to 6 weeks after self-reported COVID-19 symptom resolution except blood was collected for 1 patient after 9 weeks. Age, sex, medications, and medical problems were abstracted from UW Health's electronic medical record (EMR). Race and ethnicity were self-reported. Hospitalization and intubation for COVID-19 and smoking status at the time of blood collection (controls) or COVID-19 were obtained by EMR abstraction and self-report and were in complete agreement. Two-thirds of COVID-19 convalescent patients and all controls had a primary care appointment at UW Health within 2 years of the blood draw as an indicator of the completeness of the medical information. Patients and controls were considered to have an immunocompromising condition if they met any of the following criteria: immunosuppressing medications, systemic inflammatory or autoimmune disease, cancer not in remission, uncontrolled diabetes (secondary manifestations or hemoglobin A1c >7.0%), or congenital or acquired immunodeficiency. Controls and COVID-19 patients were similar in regard to demographics and health (S5 Data), and patients who were not hospitalized, were hospitalized, or were hospitalized and intubated also were compared (S5 Data). No patients or controls were current smokers.

### Peptide array sample binding

Samples were diluted 1:100 in binding buffer (0.01 M Tris-Cl (pH 7.4), 1% alkali-soluble casein, 0.05% Tween-20) and bound to arrays overnight at 4°C. After sample binding, the

arrays were washed 3× in wash buffer (1× TBS, 0.05% Tween-20), 10 minutes per wash. Primary sample binding was detected via Alexa Fluor 647-conjugated goat anti-human IgG secondary antibody (Jackson ImmunoResearch, West Grove, Pennsylvania, United States of America). The secondary antibody was diluted 1:10,000 (final concentration 0.1 ng/μl) in secondary binding buffer (1× TBS, 1% alkali-soluble casein, 0.05% Tween-20). Arrays were incubated with secondary antibody for 3 hours at room temperature, then washed 3× in wash buffer (10 minutes per wash), washed for 30 seconds in reagent-grade water, and then dried by spinning in a microcentrifuge equipped with an array holder. The fluorescent signal of the secondary antibody was detected by scanning at 635 nm at 2 μm resolution using an Innopsys 910AL microarray scanner. Scanned array images were analyzed with proprietary Nimble Therapeutics software to extract fluorescence intensity values for each peptide.

## Peptide microarray findings validation

We included sequences on the array of viruses that we expected all adult humans to be likely to have been exposed to as positive controls: 1 poliovirus strain (measuring vaccine exposure) and 7 rhinovirus strains. Any patient or control whose sera did not react to at least 1 positive control would be considered a failed run and removed from the analysis. All patients and controls in this analysis reacted to epitopes in at least 1 control strain (Fig 1, S1 Data).

## Peptide microarray data analysis

The raw fluorescence signal intensity values were $\log_2$ transformed. Clusters of fluorescence intensity of statistically unlikely magnitude, indicating array defects, were identified and removed. Local and large area spatial corrections were applied, and the median transformed intensity of the peptide replicates was determined. The resulting median data was cross-normalized using quantile normalization.

## Neutralization assay

Virus neutralization assays were performed with SARS-CoV-2/UW-001/Human/2020/Wisconsin on Vero E6/TMPRSS2 [79]. Virus (approximately 100 plaque-forming units) was incubated with the same volume of 2-fold dilutions of heat-inactivated serum for 30 minutes at 37°C. The antibody/virus mixture was added to confluent Vero E6/TMPRSS2 cells that were plated at 30,000 cells per well the day prior in 96-well plates. The cells were incubated for 3 days at 37°C and then fixed and stained with 20% methanol and crystal violet solution. Virus neutralization titers were determined as the reciprocal of the highest serum dilution that completely prevented cytopathic effects.

## Protein structures

The SARS-CoV-2 S-chimera.pdb used to make S protein structures is a chimeric structure built by Robert Kirchdoerfer using 6VYB.pdb, 5X4S.pdb, and 6LZG coordinates and filling in internal unresolved residues from known (presumably) analogous sites determined for SARS-CoV S from 6CRV.pdb. Additional unmodeled regions were generated using Modeller [80]. C-proximal HR2 regions were modeled as single helices (Phe1148-Leu1211) in Coot [81].

The data2bfactor Python script written by Robert L. Campbell, Thomas Holder, and Suguru Asai (downloaded from http://pldserver1.biochem.queensu.ca/~rlc/work/pymol/) was used to substitute peptide array data onto this structure in place of the B factor in PyMol (The PyMOL Molecular Graphics System, Version 2.0 Schrödinger, LLC) using a dark blue (low) to red (high) color scale. Data used for these visualizations were the average reactivity in the 40

COVID-19 convalescent patients, the average reactivity in the 20 naïve controls, and the difference between averages for the patients and for the controls.

## Enzyme-linked immunosorbent assays (ELISAs)

Costar 96-well high-binding plates (Corning, Corning, USA) were incubated at 4°C overnight with 5 μg/ml streptavidin (Thermo Fisher Scientific, Waltham, USA) in PBS (Corning). Plates were washed twice with PBS and incubated at room temperature for 1 hour with 0.5 mM of the following peptides (Biomatik, Kitchener, Canada) in PBS: 814-S-16 (KRSFIEDLLFNKVT LA-K-biotin), 1253-S-16 (CCKFDEDDSEPVLKGV-K-biotin), 390-N-16 (QTVTLLPAADLD DFSK-K-biotin), and 8-M-16 (ITVEELKKLLEQWNLV-K-biotin). Plates were washed thrice with wash buffer (0.2% Tween-20 in PBS), then incubated for 1 hour in blocking solution (5% nonfat dry milk in wash buffer) at room temperature, incubated overnight at 4°C with sera at 1:200 in blocking solution, washed 4 times with wash buffer, incubated for 1 hour at room temperature with mouse anti-human IgG conjugated to horse radish peroxidase (Southern Biotech, Birmingham, USA) diluted 1:5,000 in blocking solution, washed 4 times with wash buffer, and incubated with tetramethyl benzidine substrate solution (Thermo Fisher Scientific) for 5 minutes followed by 0.18 M sulfuric acid. Absorbance was read on a FilterMax F3 Multimode Microplate reader (Molecular Devices, San Jose, USA) at 450 and 562 nm. Background signal from 562 nm absorbance and wells with no peptide and no serum were subtracted. Plates were normalized using a pooled serum sample on every plate. Absorbance values of 0 were plotted as 0.0002 to allow a log scale for graphs. Samples were run in duplicate.

## Statistical analysis

Statistical analyses were performed in R (v 4.0.2) using in-house scripts. For each peptide, a $p$-value from a two-sided $t$ test with unequal variance between sets of patient and control responses were calculated and adjusted using the Benjamini–Hochberg (BH) algorithm. To determine whether the peptide was in an epitope (in SARS-CoV-2 proteins) or cross-reactive for anti-SARS-CoV-2 antibodies (in non-SARS-CoV-2 proteins), we used an adjusted $p$-value cutoff of <0.1 (based on multiple hypothesis testing correction for all 119,487 unique sequences on the array) and a fold-change of greater than or equal to 2 and grouped consecutive peptides as a represented epitope. Linear discriminant analysis leave-one-out cross validation was used to determine specificity and sensitivity on each peptide and from each epitope using the average signal of the component peptides. Pearson correlation for reactivity with neutralizing titer was calculated using each patient's or control's epitope signal and the $\log_2$ signal of the respective neutralization value.

To identify cross-reactive epitopes, we used each SARS-CoV-2 epitope sequence as a query, searched the database of proteins from the sequences in the peptide array using blastp (-word-size 2, num-targets 4,000) to find homologous sequences in the bat, pangolin, and other human CoV strains, then determined whether the average $\log_2$-normalized intensity for these sequences in patients was at least 2-fold greater than in controls with $t$ test statistics yielding adjusted $p$-values <0.1. Each blast hit was then mapped back to the corresponding probe ranges.

For correlations of reactivity with clinical severity, for each patient, the epitope signal was determined by averaging the normalized signal from the epitopes corresponding probes. Each epitope average signal response was fit using a multilinear regression model accounting for age, sex (Female, Male), immunocompromised status (Yes, No), and Charlson comorbidity index score [50] as additive. Contrasts between nonhospitalized and intubated patients were performed for each epitope with the fit models and $p$-values and $\log_2$ fold-change were determined.

The clinical and demographic characteristics of convalescent patients were compared to those of the controls using χ2 tests for categorical variables and Wilcoxon rank-sum tests for nonnormally distributed continuous measures.

Heatmaps were created using the gridtext [82] and complexheatmap [83] packages in R. Alignments for heatmaps were created using MUSCLE [84].

## Supporting information

**S1 Fig. Anti-SARS-CoV-2 antibody binding patterns do not vary with neutralizing titer.** Sera from 40 COVID-19 convalescent patients were assayed for IgG binding to the full SARS-CoV-2 proteome on a peptide microarray. B cell epitopes were defined as peptides in which patients' average $\log_2$-normalized intensity (black lines in line plots) is 2-fold greater than controls' (gray lines in line plots) and $t$ test statistics yield adjusted $p$-values <0.1; epitopes are identified by orange shading in the line plots. Data are grouped by their neutralizing titer. COVID-19, coronavirus disease 2019; IgG, immunoglobulin G; SARS-CoV-2, severe acute respiratory syndrome coronavirus 2.
(XLSX)

**S2 Fig. Alignment of epitopes in human and animal CoVs for which antibodies in sera from 40 COVID-19 convalescent patients showed apparent cross-reactive binding.** Alignments were performed in Geneious Prime 2020.1.2 (Auckland, New Zealand). CoV, coronavirus; COVID-19, coronavirus disease 2019.
(PDF)

**S1 Table. Metadata for the 40 COVID-19 convalescent patients and 20 naïve controls: Patient or control status, age at blood draw, legal sex, Charlson comorbidity index, immunocompromised status, whether the COVID-19 convalescent patients required hospitalization or intubation for their COVID-19 course, race, and ethnicity.** Neutralizing titers and the dates of the first positive COVID-19 test are also included. COVID-19, coronavirus disease 2019.
(XLSX)

**S2 Table. Specificity and sensitivity for past SARS-CoV-2 infection in 40 COVID-19 convalescent patients compared to 20 naïve controls of individual 16-mer peptides comprising epitopes throughout the full SARS-CoV-2 proteome.** COVID-19, coronavirus disease 2019; SARS-CoV-2, severe acute respiratory syndrome coronavirus 2.
(XLSX)

**S3 Table. Epitopes paired with the 1-M-24 epitope obtained AUC-ROC of 1.0 for SARS-CoV-2 infection in 40 COVID-19 convalescent patients and 20 naïve controls using leave-one-out cross validation with linear discriminant analysis.** AUC-ROC, area under the receiver operating characteristic curve; COVID-19, coronavirus disease 2019; SARS-CoV-2, severe acute respiratory syndrome coronavirus 2.
(XLSX)

**S4 Table. Comparison of antibody binding in SARS-CoV-2 B cell epitopes in 8 intubated COVID-19 convalescent patients compared to 25 symptomatic but never hospitalized COVID-19 convalescent patients compared by multilinear regression accounting for age, sex, immunocompromising conditions, and Charlson comorbidity index score.** COVID-19, coronavirus disease 2019; SARS-CoV-2, severe acute respiratory syndrome coronavirus 2.
(XLSX)

**S5 Table. B cell epitopes detected by sera from 40 individuals having a first positive SARS-CoV-2 test in March through May 2020 overlap with protein areas containing in SARS-CoV-2 variants of concern.** SARS-CoV-2, severe acute respiratory syndrome coronavirus 2. (XLSX)

**S1 Data. Percentages and individual reactivity of the 40 COVID-19 convalescent patients and 20 naïve controls reacted to known epitopes in at least 1 control virus (rhinovirus and poliovirus strains).** COVID-19, coronavirus disease 2019. (XLSX)

**S2 Data. Percentages and individual data for the 40 COVID-19 convalescent patients and 20 naïve controls showing log$_2$-normalized fluorescence intensity at least 3.00 standard deviations above the mean for the array for 9 species of CoVs.** CoV, coronavirus; COVID-19, coronavirus disease 2019. (XLSX)

**S3 Data. Cross-reactive binding of antibodies against other CoVs in 40 COVID-19 convalescent patients compared to 20 naïve controls.** CoV, coronavirus; COVID-19, coronavirus disease 2019. (XLSX)

**S4 Data. Cross-reactive binding of antibodies in 40 COVID-19 convalescent patients compared to 20 naïve controls in protein motifs in other CoVs aligned to SARS-CoV-2.** CoV, coronavirus; COVID-19, coronavirus disease 2019; SARS-CoV-2, severe acute respiratory syndrome coronavirus 2. (XLSX)

**S5 Data. Supporting information tables (3 tables) show data relevant to the Methods for this study.** Supporting information Table A contains the proteins represented on the array, including the GenBank accession numbers and the number of replicates of reach peptide in those proteins. Supporting information Table B contains the characteristics of the 40 COVID-19 convalescent patients and the 20 naïve controls whose sera were used in this study. Supporting information Table C contains the characteristics of the 40 COVID-19 convalescent patients according to hospitalization status. COVID-19, coronavirus disease 2019. (DOCX)

## Acknowledgments

The authors are grateful to Mr. Eric Sullivan, Dr. Richard Pinapati, Dr. John Tan, Dr. Daniel Agnew, Dr. Brad Garcia, and Dr. Jigar Patel, all of Nimble Therapeutics for access to their unique peptide array technology. The authors are also grateful to Dr. Christina Newman, Dr. Nathan Sherer, Dr. Thomas Friedrich, Dr. Amelia Haj, Dr. James Gern, Dr. Christine Seroogy, and Gage Moreno for their thoughtful comments and helpful discussions in preparing this manuscript. The authors also thank Dr. Robert Kirchdoerfer for generously providing the chimeric pdb file used for structural representations of SARS-CoV-2 spike protein in this work (see Fig 4 and "Protein structures" in the Methods).

## Author Contributions

**Conceptualization:** Anna S. Heffron, Sean J. McIlwain, Maya F. Amjadi, David A. Baker, Yoshihiro Kawaoka, Miriam A. Shelef, David H. O'Connor, Irene M. Ong.

**Data curation:** Anna S. Heffron, Sean J. McIlwain, Maya F. Amjadi, David A. Baker, Saniya Khullar, Tammy Armbrust, Peter J. Halfmann, Miriam A. Shelef, Irene M. Ong.

**Formal analysis:** Anna S. Heffron, Sean J. McIlwain, Maya F. Amjadi, David A. Baker, Tammy Armbrust, Peter J. Halfmann, Ajay K. Sethi, Ann C. Palmenberg, Miriam A. Shelef, Irene M. Ong.

**Funding acquisition:** Anna S. Heffron, Maya F. Amjadi, Yoshihiro Kawaoka, Miriam A. Shelef, David H. O'Connor, Irene M. Ong.

**Investigation:** Anna S. Heffron, Sean J. McIlwain, Maya F. Amjadi, David A. Baker, Tammy Armbrust, Peter J. Halfmann, Yoshihiro Kawaoka, Miriam A. Shelef, David H. O'Connor, Irene M. Ong.

**Methodology:** Anna S. Heffron, Sean J. McIlwain, Maya F. Amjadi, David A. Baker, Ann C. Palmenberg, Miriam A. Shelef, Irene M. Ong.

**Project administration:** Anna S. Heffron, Miriam A. Shelef, Irene M. Ong.

**Resources:** Yoshihiro Kawaoka, Miriam A. Shelef, David H. O'Connor, Irene M. Ong.

**Software:** Anna S. Heffron, Sean J. McIlwain, David A. Baker, Saniya Khullar, Ann C. Palmenberg, Irene M. Ong.

**Supervision:** Anna S. Heffron, Yoshihiro Kawaoka, Ann C. Palmenberg, Miriam A. Shelef, David H. O'Connor, Irene M. Ong.

**Validation:** Anna S. Heffron, Sean J. McIlwain, Maya F. Amjadi, David A. Baker, Saniya Khullar, Tammy Armbrust, Peter J. Halfmann, Miriam A. Shelef, Irene M. Ong.

**Visualization:** Anna S. Heffron, Sean J. McIlwain, Maya F. Amjadi, David A. Baker, Saniya Khullar, Ann C. Palmenberg, Miriam A. Shelef, David H. O'Connor, Irene M. Ong.

**Writing – original draft:** Anna S. Heffron, Sean J. McIlwain, Maya F. Amjadi, Peter J. Halfmann, Irene M. Ong.

**Writing – review & editing:** Anna S. Heffron, Sean J. McIlwain, Maya F. Amjadi, David A. Baker, Saniya Khullar, Tammy Armbrust, Peter J. Halfmann, Yoshihiro Kawaoka, Ajay K. Sethi, Ann C. Palmenberg, Miriam A. Shelef, David H. O'Connor, Irene M. Ong.

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
