## [Editor Report · Decision Letter 0]

18 Jan 2021

Dear Dr. Ong, 

Thank you for submitting your manuscript entitled "The landscape of antibody binding in SARS-CoV-2 infection" for consideration as a Research Article by PLOS Biology.

Your manuscript has now been evaluated by the PLOS Biology editorial staff, as well as by an academic editor with relevant expertise, and I am writing to let you know that we would like to send your submission out for external peer review.

Please re-submit your manuscript within two working days, i.e. by Jan 20 2021 11:59PM.

Kind regards,

Paula

---

Associate Editor

PLOS Biology

---

## [Decision Letter · Decision Letter 1]

19 Mar 2021

Dear Dr. Ong,

Thank you very much for submitting your manuscript "The landscape of antibody binding in SARS-CoV-2 infection" for consideration as a Research Article at PLOS Biology. Your manuscript has been evaluated by the PLOS Biology editors, an Academic Editor with relevant expertise, and by several independent reviewers.

In light of the reviews (below), we are pleased to offer you the opportunity to address the comments from the reviewers in a revised version that we anticipate should not take you very long. We will then assess your revised manuscript and your response to the reviewers' comments and we may consult the reviewers again. Please also make sure to address the following data and other policy-related requests.

In particular, reviewer #1 thinks that the manuscript would be strengthened if you could comment on whether the non-homologous, non-identical epitopes recognized by SARS-CoV-2 positive sera in other CoVs are conserved within themselves and comment on what potentially may be inducing these specific responses, assess whether sera recognized variant/mutated versions of the S peptides for the B1.1.7, P.1, and B.1.351 variants, a more thorough description of the 9 epitopes that appeared to correlate with disease severity epitopes would be informative, and asks whether there were any epitopes found that correlated with decreased disease severity. This reviewer also says that to conclude that the ACE2 binding site is not “immunodominant” you need to include conformational epitopes, RBD and/or Spike proteins (not peptides derived from them) for comparison on the microarray as necessary controls, however, we consider that this is beyond the scope of this technology and this won't be necessary for publication. Reviewer #2 thinks that you should include information of sample collecting time post disease onset, provide neutralizing titres of the sera from COVID-19 patients and controls, and says that it will be of great interest to determine whether the epitopes identified in the study could elicit neutralizing antibodies after vaccination.

Please also address the following data and other policy-related requests.

DATA POLICY:

Regardless of the method selected, please ensure that you provide the individual numerical values that underlie the summary data displayed in the following figure panels as they are essential for readers to assess your analysis and to reproduce it: Figure 6A, 6B and 7.

**Please also ensure that figure legends in your manuscript include information on where the underlying data can be found, **and ensure your supplemental data file/s has a legend.

Please clarify the Conflict of Interest statement as you say in our system that the authors declare no competing interest but in the manuscript you declare authors to be listed as inventors on a patent related to the study. 

We expect to receive your revised manuscript within 1 month.

**IMPORTANT - SUBMITTING YOUR REVISION**

*Resubmission Checklist*

*Published Peer Review*

*PLOS Data Policy*

*Blot and Gel Data Policy*

Sincerely,

Paula

---

Associate Editor,

pjaureguionieva@plos.org,

PLOS Biology

REVIEWS:

Reviewer #1: Immunity against virus, structural biologist, immunogen engineering.

Reviewer #2: Immune response against coronavirus.

Reviewer #1: In this manuscript, Heffron et al. report linear peptide binding antibodies from pre-pandemic and SARS-CoV-2 infected patient serum samples to the S, M, N, ORF1ab, ORF3a, ORF6, and ORF8 proteins of SARS-CoV-2 as well as to proteins in the common cold CoVs HKU1, OC43, NL63, and 229E. The extent of this binding was measured first by peptide microarray and then validated by ELISA with selected peptides of interest. Several studies to date have performed similar types of analyses characterizing linear epitopes recognized by convalescent SARS-CoV-2 sera to SARS-CoV-2 proteins as well as other CCCoV proteins thus this limits the overall impact of the present study. Furthermore, the rather cursory and brief analysis of the data (in both the results section and discussion) does disservice to the work itself and ultimately fails to convey to the reader the overall importance.

The manuscript would be strengthened if the authors could comment on:

- Whether the non-homologous, non-identical epitopes recognized by SARS-CoV-2 positive sera in RaTG13, pangolin CoV, and SARS-CoV are conserved within themselves and comment on what potentially may be inducing these specific responses.

- The authors mention that sera can recognize B1.1.7 ORF8 and N binding—is that notable—as aren't these regions conserved between B1.1.7 and SARS-CoV-2? Additionally, it would be more informative and noteworthy to assess whether sera recognized variant/mutated versions of the S peptides for the B1.1.7, P.1, and B.1.351 variants.

- The observation that 9 epitopes appeared to correlate with disease severity is interesting. While the data is presented in Figure 7 a more thorough description of these epitopes would be informative. Additionally, were any epitopes found that correlated with decreased disease severity?

- In the Discussion, the authors comment "The ACE2 binding site and S-helix in extended fusion are not as immunodominant as expected suggesting that other, less-investigated may be playing a larger role in immunity to SARS-CoV-2…" is problematic. Arguably, the vast majority of the elicited response to vaccination or infection is conformational specific. Using linear peptides, as done in this microarray, will not "capture" these responses. For the authors to suggest that ACE2 binding site, which is, in fact, a conformational specific epitope recognized, predominantly by conformational specific antibodies (e.g., B38, ADI-) is not "immunodominant" in their assay is misleading and inaccurate. Inclusion of RBD and/or Spike proteins (not peptides derived from them) for comparison on the microarray are necessary controls. Furthermore, it is unclear what the authors mean by "S-helix in extended fusion".

Minor:

- Discussion paragraph 2: "though that may be due to using an earlier sample" is ambiguous phrasing leaving it unclear which sample (anti-M positive or negative) is earlier.

- Discussion paragraph 4: "especially given that pre-existing anti-CoV antibodies are more common in children and adolescents" implies a connection between protection and age that is not clearly stated in the text and should be clarified.

- In the Acknowledgements what is meant by "chimeric PDB file" used for Figure 4

Figures

- Figures 2 and 3: the black arrows are not defined in the figure legends.

- Figure 4: a scale bar for the heatmap should be included.

- Figure 4: use of a rotated arrow versus a linear one that denotes a 45deg rotation should be used

- Figure 4: the authors use the "one up RBD" structure; it would be useful to readers who are not familiar with the S structure to include a non-heat map version that labels key regions (e.g., RBD, RBM, fusion peptide, "base").

Reviewer #2: In this manuscript, Heffron and colleagues designed a peptide microarray of SARS-CoV-2 and other coronaviruses to assess antibody epitope specificity and potential cross-reactivity with other CoVs in COVID-19 convalescent patients and demonstrated previously unknown, highly reactive B cell epitopes throughout the full proteome of SARS-CoV-2 and other CoV proteins. An epitope in the N-terminus of M protein with high specificity and sensitivity to the serum of the COVID-19 patients was found. This study will be of useful for vaccine design and serological diagnosis. This reviewer just has a few concerns.

1. Humoral immunity to virus is related to infection time. It suggested that information of sample collecting time post disease onset should be included somewhere in the manuscript.

2. Neutralizing titers of the sera from COVID-19 patients and the controls should be provided. The control sera showed cross reactivity with the SARS-CoV-2 peptide microarray. Did the control sera neutralize SARS-CoV-2?

3. It will be of greater interest to determine whether epitopes identified in this study could elicit neutralizing antibodies after vaccination.

---

## [Editor Report · Decision Letter 2]

6 May 2021

Dear Dr. Ong,

On behalf of my colleagues and the Academic Editor, Galit Alter, I am pleased to say that we can in principle offer to publish your Research Article "The landscape of antibody binding in SARS-CoV-2 infection" in PLOS Biology, provided you address any remaining formatting and reporting issues. These will be detailed in an email that will follow this letter and that you will usually receive within 2-3 business days, during which time no action is required from you. Please note that we will not be able to formally accept your manuscript and schedule it for publication until you have made the required changes.

PRESS

Thank you again for supporting Open Access publishing. We look forward to publishing your paper in PLOS Biology. 

Sincerely, 

Paula

---

Paula Jauregui, PhD 

Associate Editor 

PLOS Biology